# EDNRA-Expressing Mesenchymal Cells Are Expanded in Myeloma Interstitial Bone Marrow and Associated with Disease Progression

**DOI:** 10.3390/cancers15184519

**Published:** 2023-09-12

**Authors:** Wen Ling, Sarah K Johnson, Syed J Mehdi, Daisy V Alapat, Michael Bauer, Maurizio Zangari, Carolina Schinke, Sharmilan Thanendrarajan, Frits van Rhee, Shmuel Yaccoby

**Affiliations:** Myeloma Center, Department of Internal Medicine, Winthrop P. Rockefeller Cancer Institute, University of Arkansas for Medical Sciences, Little Rock 72205, AR, USA; wling@uams.edu (W.L.); skjohnson@uams.edu (S.K.J.); sjmehdi@uams.edu (S.J.M.); dvalapat@uams.edu (D.V.A.); mbauer2@uams.edu (M.B.); mzangari@uams.edu (M.Z.); cdschinke@uams.edu (C.S.); sthanendrarajan@uams.edu (S.T.); vanrheefrits@uams.edu (F.v.R.)

**Keywords:** myeloma, pericytes, angiogenesis, mesenchymal cells, microenvironment

## Abstract

**Simple Summary:**

In multiple myeloma, malignant plasma cells accumulate in the bone marrow. The symptoms and complications include osteolytic bone disease, anemia, reduced kidney function, immunosuppression, and induction of tumor-associated angiogenesis. Mesenchymal cells are important components of bone marrow niches; studying their dysfunctional activities reveals key features in myeloma pathogenies and improves therapies. We discovered that a subset of mesenchymal cells that express the receptor EDNRA are more prevalent in bone marrow areas infiltrated with tumor cells. In normal conditions, these cells are attached to blood vessels and mediate their functions; but in myeloma they seem to detach and accumulate in the interstitial marrow. The proportion of EDNRA-expressing cells and the expression of EDNRA in bone biopsies is low in premalignant stages and highest in high-risk myeloma patients. EDNRA could serve as a useful novel clinical biomarker of disease progression and dysfunctional bone marrow vasculature.

**Abstract:**

Multiple myeloma (MM) induces dysfunctional bone marrow (BM) mesenchymal cells and neoangiogenesis. Pericytes and smooth muscle cells (SMCs) could detach from vessels and become cancer-associated fibroblasts. We found that the pericyte and SMC marker endothelin receptor type A (EDNRA) is overexpressed in whole MM bone biopsies; we sought to characterize its expression. EDNRA expression gradually increased with disease progression. High-risk MM patients had higher EDNRA expression than low-risk MM patients and EDNRA expression was highest in focal lesions. High EDNRA expression was associated with high expression of pericyte markers (e.g., RGS5, POSTN, and CD146) and the angiogenic marker FLT1. A single-cell analysis of unexpanded BM mesenchymal cells detected EDNRA expression in a subset of cells that coexpressed mesenchymal cell markers and had higher expression of proliferation genes. Immunohistochemistry revealed that the number of EDNRA+ cells in the interstitial BM increased as MM progressed; EDNRA+ cells were prevalent in areas near the MM focal growth. EDNRA+ cells were detached from CD34+ angiogenic cells and coexpressed RGS5 and periostin. Therefore, they likely originated from pericytes or SMCs. These findings identify a novel microenvironmental biomarker in MM and suggest that the presence of detached EDNRA+ cells indicates disrupted vasculature and increased angiogenesis.

## 1. Introduction

Multiple myeloma (MM) develops from a premalignant stage called MGUS (monoclonal gammopathy of undetermined significance) [1]. MGUS progresses through multiple steps to become smoldering MM (SMM), which then progresses to low-risk MM and eventually to high-risk MM with an extramedullary growth pattern [1]. The progression of MM is associated with changes in the tumor microenvironment during each stage of the disease; the roles of these changes in MM disease progression is a matter of continual research [2,3,4,5].

The bone marrow (BM) mesenchymal stem cells (MSCs) and their lineages are affected by MM; the cellular and molecular changes that occur in BM because of MM are only partially known. A study of BM cellularity with single-cell RNA sequencing (scRNA-seq) indicated that the mesenchymal compartment is molecularly and phenotypically heterogeneous [6]. In MM, both the number of osteoblasts [7,8] and the number of adipocytes [9] are increased at the initial stages of disease; as MM progresses, their numbers decrease in focal lesions (FLs) and interstitial BM. These changes occurred in part because MM suppressed MSC differentiation and induced senescence in MSCs [10].

Gene expression profiling (GEP) [10,11,12] and scRNA-seq [13] were used to identify mesenchymal cell populations that are altered in MM. Using GEP of whole bone biopsies from patients with MGUS, SMM, or MM, we studied the gene signature of the BM mesenchymal compartment based on a set of genes highly expressed by MSCs and developed a BM biopsy-based three-gene score that is associated with disease progression and outcome [12]. The three genes include *COL4A1*, which is known to be overexpressed and associated with angiogenesis, and *NPR3* and *ITGBL1*, which are under expressed in whole BM biopsy samples from patients with MM [12]. We also detected a unique population of small adipocytes in red marrow; these cells were diminished in BM areas highly infiltrated with MM cells [10]. A different group recently used scRNA-seq to detect a rare population of inflammatory MSCs; this population was not evident in normal BM [13].

The other major nonhematopoietic compartment in BM is the vasculature. The pericytes and smooth muscle cells (SMCs), often referred as mural cells, play an essential role in the development of the vasculature. They promote vascular quiescence and long-term vessel stabilization and modulate inflammation through their interactions with endothelial cells [14,15]. Studies in solid tumors showed that mural cells could detach from established vessels and transition to become cancer-associated fibroblasts (CAFs) [16]. As in many malignancies, the development of MM in BM is associated with increased neoangiogenic micro vessels which lack SMCs and pericytes [17]. Using our GEP database, we identified endothelin receptor type A (*EDNRA*) as a microenvironmental marker that is overexpressed in bone biopsies of patients with MM. Endothelins play an essential role in controlling vasoconstriction and the inflammatory processes produced by vascular endothelial cells (VECs) through the expression of EDNRA on the surface of pericytes and SMCs [18,19,20]. In the present study, we shed light on the cells that express this marker in the MM BM milieu.

## 2. Materials and Methods

### 2.1. Bone Biopsy Samples

All protocols were approved by the Institutional Review Board at the University of Arkansas for Medical Sciences (UAMS) and clinical results were reported annually. Specimens were obtained from random BM of the iliac crest or from CT-guided fine-needle biopsies of MRI-defined FLs of patients enrolled in our TT2-TT5 Total Therapy clinical trials (registration numbers NCT00083551, NCT00081939, NCT00572169, NCT00734877, NCT00869232) before initiation of treatment. Details of these clinical trials have been reported [21,22]. These samples were used for GEP and immunohistochemistry analyses. BM biopsy samples from healthy donors were obtained from random BM of the iliac crest and used for GEP analysis^12^; samples obtained from femur heads of age-matched individuals who underwent orthopedic surgery at UAMS were used for immunohistochemistry [10].

### 2.2. Processing Samples for Gene Expression Analysis

The biopsy samples used to generate the GEP data are available at GEO DataSets (GSE136324 and GSE136337) and data sets are available and described elsewhere [11]. RNA isolation and GEP were performed with the Human Genome U133 Plus 2.0 Array (Affymetrix, Santa Clara, CA, USA), as previously described [23,24]. GEP analyses on MM cell lines and on primary MM cells, which had been isolated from BM aspirates by CD138 immunomagnetic bead selection, and the GEP-based scores, used to define low-risk (LR) and high-risk (HR) MM, were previously described [23,24]. In most cases, GEP analyses were performed on whole bone biopsies and purified CD138+ MM PC from the same patients [11]. GEP data were available for baseline BM biopsy samples from patients with MGUS (*n* = 55), SMM (*n* = 76), or newly diagnosed MM patients (*n* = 354) [11]. Also, biopsy samples from healthy donors (*n* = 68) and paired samples (i.e., from the same MM patient) of random BM and FL biopsy (*n* = 49) were available. 

### 2.3. Single- and Double-Staining Immunohistochemistry

EDNRA and CD34 antibodies were purchased from Abcam (Waltham, MA, USA). Periostin and RGS5 antibodies were purchased from LSBio (Shirely, MA, USA). Ig kappa and lambda antibodies were purchased from ThermoFisher Scientific (Waltham, MA, USA). Immunohistochemistry was performed on bone biopsy sections as described [10]. Briefly, after peroxidase quenching (3% hydrogen peroxide, 10 min), slides were incubated with EDNRA antibody. Assays were completed with the Dako LSAB2 system-HRP kit (Agilent Pathology Solutions, Santa Clara, CA, USA) and counterstained with hematoxylin. For EDNRA, CD34 and Ig kappa or lambda double staining, the Multiview (mouse HRP/mouse AP) IHC kit (Enzo Life Sciences, Farmingdale, NY, USA) was used according to the manufacturer’s instructions. For sequential staining of EDNRA and CD34, RGS5, or periostin, the double staining IHC kit (HRP/Green and Fast Red) was used (Abcam, Cambridge, MA, USA). To avoid false positives, the counterstaining step with hematoxylin was omitted. An Olympus BH2 microscope (Olympus, Melville, NY, USA) was used to obtain images with a SPOT 2 digital camera (Diagnostic Instruments Inc, Sterling Heights, MI, USA). Adobe Photoshop version 10 (Adobe Systems, San Jose, CA, USA) was used to process the images.

### 2.4. Single-Cell Analysis of Unexpanded Mesenchymal Cell

Unexpanded mesenchymal cells were isolated and sorted as described [10]. To analyze gene expression, we used qRT-PCR to amplify 34 genes. These genes included mesenchymal cell genes overexpressed or underexpressed in myelomatous bones [10,12] They also included housekeeping genes, hematopoietic cell markers, MSC markers, proliferation and cell cycle-related genes, differentiation-associated markers, growth factors, chemokines, and the adhesion molecules [10].

## 3. Statistical Analyses

For *EDNRA* gene expression identified by GEP, multiple comparison adjustments were used to calculate the *q*-value [11,24]. The values of experimental data are expressed as mean ± SEM. For quantification of the proportion of EDNRA+ cells in the interstitial BM, EDNRA+ cells from five nonoverlapping areas in 20× photographs were counted and the unpaired Student’s *t* test was used to determine the significance of *p* < 0.05 [10]. For evaluating gene expression in single cells, Student’s *t* test was used to determine the significance of *p* < 0.001.

## 4. Results

### 4.1. EDNRA Is Overexpressed in the MM BM Microenvironment and Is Associated with Disease Progression and Poor Outcome

Expression of *EDNRA* was analyzed based on GEP of whole bone biopsies as described [11]. *EDNRA* gene expression was lower in MM cell lines and in primary MM plasma cells than in whole bone biopsies. In biopsy samples, *EDNRA* expression was lower in normal donors. As MM progressed from premalignant stages (i.e., MGUS and SMM) to newly diagnosed MM, *EDNRA* expression gradually increased (Figure 1A). In bone biopsies from newly diagnosed MM patients, *EDNRA* expression was higher in the random BM of HR MM patients than in the random BM of LR MM patients. Overall, *EDNRA* expression was highest in FLs (Figure 1A). In a set of paired random BM and FL samples, we examined *EDNRA* expression, along with expression of *RGS5* and *MCAM* (CD146), which are associated with pericytes and SMCs, and with *FLT1*, which is associated with angiogenesis. We found that for all four genes, expression was higher in FLs than random BM of the same patients (Figure 1B).

We analyzed the outcome of newly diagnosed MM patients who were enrolled in the TT3a clinical trial [22] and who had whole biopsy GEP performed at diagnosis. A higher expression of EDNRA was significantly associated with shorter progression-free survival and overall survival (Figure 2). These findings indicate that *EDNRA* is upregulated in the BM microenvironment of patients with MM and that expression is associated with disease progression and poor outcomes.

### 4.2. In Interstitial BM of Patients with MM, EDNRA Is Detected in Vascular SMCs and in Cells That Resemble Fibroblasts

To identify the cell types that express EDNRA in BM, we performed IHC staining for EDNRA on bone sections from healthy donors and patients with MGUS, SMM, or MM. In bones from healthy donors, EDNRA was detected in SMCs covering mature vessels (Figure 3A) but was not detected in the interstitial BM. SMCs of patients with MGUS, SMM, or MM also expressed EDNRA in mature vessels of bone sections (Figure 3B).

Double staining of bone sections for EDNRA and CD34 detected VECs in the inner part of the mature vessel, covered by layers of EDNRA+ SMCs (Figure 3C). Further observations revealed that individual EDNRA+ cells were present in interstitial BM of patients with MM and that MM cells were negative for this marker. EDNRA+ cells had a fibroblast-like morphology, which suggests they were of mesenchymal origin (Figure 3D). EDNRA and CD34 double staining revealed that EDNRA+ mesenchymal cells were dispersed throughout the BM of MM patients and were separate from CD34 angiogenic vessels (Figure 3E). These findings suggest that although EDNRA expression along mature blood vessels can be found in normal and MM conditions, EDNRA+ cells are also found in the interstitial marrow of patients with MM and these cells are distinct mesenchymal cells.

### 4.3. Frequency of EDNRA+ Cells in Interstitial BM Increases with MM Disease Stage

To test whether EDNRA+ cells were found more frequently in MM samples, we performed IHC for EDNRA to detect EDNRA+ cells in BM samples from patients with MGUS, SMM, LR MM, or HR MM. We then quantified the number of EDNRA+ mesenchymal cells in the interstitial BM of each sample. The number of EDNRA+ mesenchymal cells was 5.3-fold higher in patients with LR MM than in patients with MGUS or SMM (*p* < 0.01) and was 2.2-fold higher in HR MM patients than in LR MM patients (*p* < 0.03, Figure 4). In patients with LR MM, EDNRA+ cells were more evident in highly involved areas and were absent in uninvolved interstitial BM (Figure 5A). Double staining for EDNRA and clonal immunoglobulin light chain expressed by MM cells revealed EDNRA+ mesenchymal cells near MM cells (Figure 5B).

### 4.4. EDNRA+ Cells Coexpress the Pericyte Markers RGS5 and Periostin

EDNRA is a marker of pericytes and is often detected in SMCs [18]. To further explore the phenotype of EDNRA+ cells in interstitial BM, we double stained bone sections from MM patients for EDNRA and two markers associated with pericytes, RGS5, and periostin [6]. Double staining for EDNRA and periostin revealed that these two markers were coexpressed in fibroblast-like mesenchymal cells. In contrast, periostin, but not EDNRA, was detected in BM adipocytes and hematopoietic cells were negative for both markers (Figure 6A,B).

Double staining for EDNRA and RGS5 revealed coexpression in mesenchymal cells but not in other cell types in the interstitial BM of patients with MM (Figure 6C,D).

### 4.5. Unexpanded Single Mesenchymal Cells Express EDNRA

To shed light on the characteristics of BM *EDNRA*-expressing cells, we sorted individual unexpanded mesenchymal cells from healthy donors and patients with MM. Based on the collagenase enzymatic digestion method [10], we predicted that the isolated cells included mesenchymal cells that were detached from established vasculature and bone surfaces. We used qRT-PCR to measure expression of selected genes associated with MSCs and mesenchymal cells. We also analyzed expression of *MKI67*, *CCND1*, *CDKN1A*, and *CDKN2A* (p16) as markers of cellular proliferation and senescence, and we used expression of *PTPRC* (CD45) and *CD34* to exclude cells of hematopoietic and endothelial origins.

Overall, we combined the cells from both samples and analyzed 91 mesenchymal cells. These cells expressed typical mesenchymal cell genes, such as *FAP*, *FN1*, *COL1A1*, *CXCL12*, and *CD44* (Figure 7). We compared expression of genes in cells that express high levels of *EDNRA* (*n* = 31) to cells that either did not express *EDNRA* or expressed low levels of this gene (*n* = 60). *EDNRA*-expressing cells expressed significantly higher levels of *COL4A1*, *COL4A2*, *CSPGS*, and *CYR61*; all are closely associated with SMCs and pericytes. Also, EDNRA+ cells expressed high levels of the proliferation genes *CCND1* and *MKI67* (Figure 7). Expression of the mesenchymal stem cell associated genes *ALCAM*, *FOXC1,* and *LEPR* were similar, whereas expression of *IGF2*, but not *IGF1* or *TGFB1*, was higher in EDNRA+ cells. The chemokine receptor *CXCL12* was moderately higher in EDNRA+ cells, whereas we detected similar expression of *CD44* between the two subsets of cells. Overall, these findings indicate that EDNRA+ cells are a type of mesenchymal cell and that these cells express genes that are related to the vasculature.

## 5. Discussion

EDNRA is a recognizable pericyte marker often detected on SMCs that cover the vasculature; it is not detected in ECs [6,18,20]. Earlier studies showed that *EDNRA* is also expressed by osteoblasts and that the Wnt signaling inhibitor DKK1 blocked endothelin 1-induced osteoblast differentiation and bone formation [25,26]. Although previous studies detected EDNRA expression in MM cells [27], our GEP data indicated lowest expression in CD138+ purified plasma cells and in MM cell lines; the immunohistochemistry staining of patients’ biopsies showed no expression of this protein in malignant MM cells. In our study, we identified EDNRA+ cells and characterized them as a type of mesenchymal cell expanded in the interstitial BM of patients with MM and detached from the vasculature and bone surface. These cells resembled pericytes or SMCs as they coexpressed RGS5 and periostin. We further showed that these cells are more evident in BM areas infiltrated with MM cells and they reside alongside neoangiogenic ECs, which are known to form in MM-infiltrated BM areas [17]. *EDNRA* gene expression in whole bone biopsies and the frequency of EDNRA+ cells were closely associated: both of these parameters increased with disease stage and both *EDNRA* expression and the number of EDNRA+ cells were highest in FLs and in the interstitial BM of patients with HR MM. Single-cell analyses of unexpanded BM mesenchymal cells detected EDNRA+ cells that coexpressed mesenchymal cell matrix factors (e.g., *FN1*, *COL4A1*, *COL4A2*) and markers of MSCs (e.g., *FAP*, *FOXC1*, *LEPR*). Additionally, EDNRA+ mesenchymal cells expressed higher levels of the proliferating markers *MI67* and *CCND1* than mesenchymal cells that did not express *EDNRA*. Notably, *COL4A1* is one of three MSC genes constituting a BM biopsy-based three-gene score that is associated with disease progression and overall survival of patients with MM [12].

In solid tumors, vascular pericytes and SMCs can detach from the microvasculature and transition to become stromal fibroblasts resembling CAFs [16]. The separation of pericytes from the BM vasculature has not been reported in MM, though such a scenario may exist in this disease. First, ANGPT1 and ANGPT2 are two secreted factors with contrasting effects on pericyte attachment to the vasculature. As MM progresses, their ratio, which reflects vessel stabilization, is significantly lower; this low ratio is associated with increased angiogenesis [28]. Also, there is a positive correlation between the ANGPT1/ANGPT2 ratio and the A parameter measured by dynamic contrast-enhanced MRI (DCE-MRI) [29]. The DCE-MRI amplitude A parameter, which reflects blood volume, can predict response and outcome in patients with MM [30]. Interestingly, thalidomide, a frontline anti-MM clinical drug known to inhibit angiogenesis, induces vessel maturation and stability by acting on pericytes and increasing mural cell coverage of the vasculature [31]. The mechanisms by which MM promotes the expansion of EDNRA+ cells in the interstitial BM require further investigation.

## 6. Conclusions

In summary, our findings suggest that in MM, the disruption of the normal vascularization of BM permits neoangiogenesis and expansion of mesenchymal cells, such as detached EDNRA+ mural cells, all of which may support MM progression. Although further investigation is needed to better understand these mechanisms, our studies show that *EDNRA* expression and the proportion of EDNRA+ cells in the interstitial myelomatous BM can serve as biomarkers of disease progression in MM. Finally, we show that these cells, which are of a mesenchymal origin, are previously unrecognized players in the pathogenesis of MM. Together, this information furthers our understanding of the molecular and phenotypic changes that occur in the MM BM microenvironment. 

## Figures and Tables

**Figure 1 cancers-15-04519-f001:**
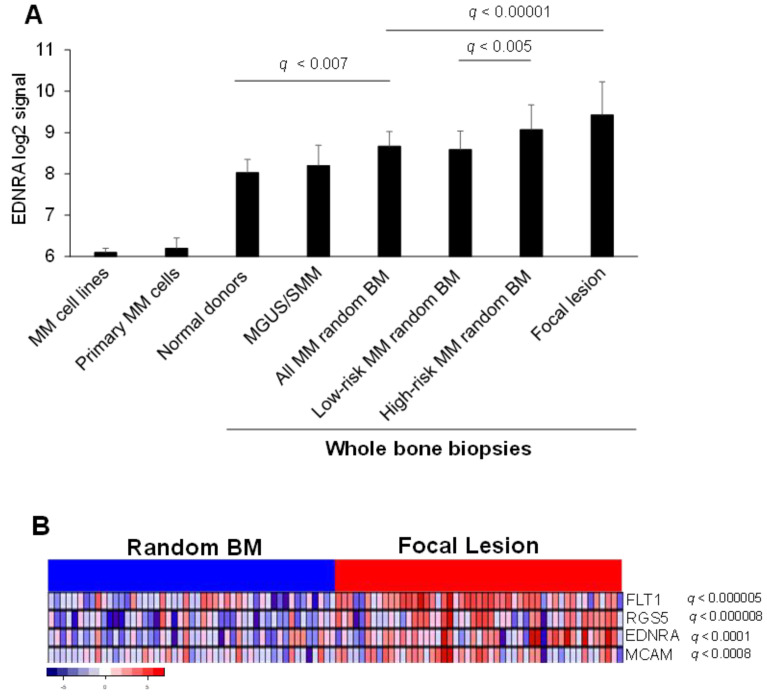
*EDNRA* is overexpressed in the MM microenvironment and linked to angiogenesis. (**A**) EDNRA signal intensity in MM cell lines, CD138-selected MM plasma cells from newly diagnosed MM patients (Primary MM cells) and in whole bone biopsies of normal donors and of MGUS/SMM, newly diagnosed MM patients, newly diagnosed MM patients segregated based on GEP risk score (i.e., low risk or high risk), and focal lesion. (**B**) Expression of *EDNRA*, *RGS5*, *MCAM* (CD146), and *FLT1* in paired random bone biopsies and focal lesion of newly diagnosed MM patients. Data expressed as mean ± SEM.

**Figure 2 cancers-15-04519-f002:**
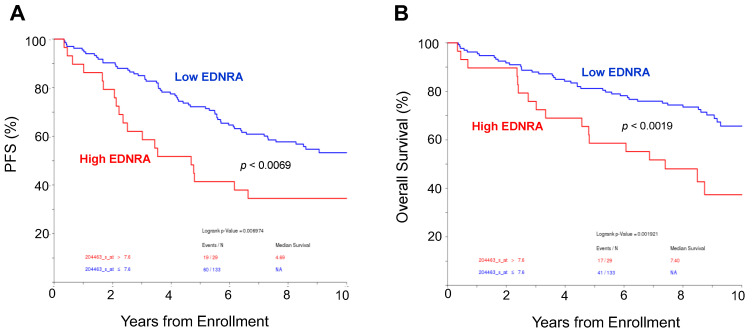
High expression of *EDNRA* in bone biopsies is associated with poor outcome. (**A**,**B**). Progression-free survival (PFS, **A**) and overall survival (**B**) of newly diagnosed MM patients enrolled in the TT3A clinical trial, based on high or low expression of *EDNRA* in whole bone biopsy at diagnosis. Outcome is shown for the 10 years since enrollment.

**Figure 3 cancers-15-04519-f003:**
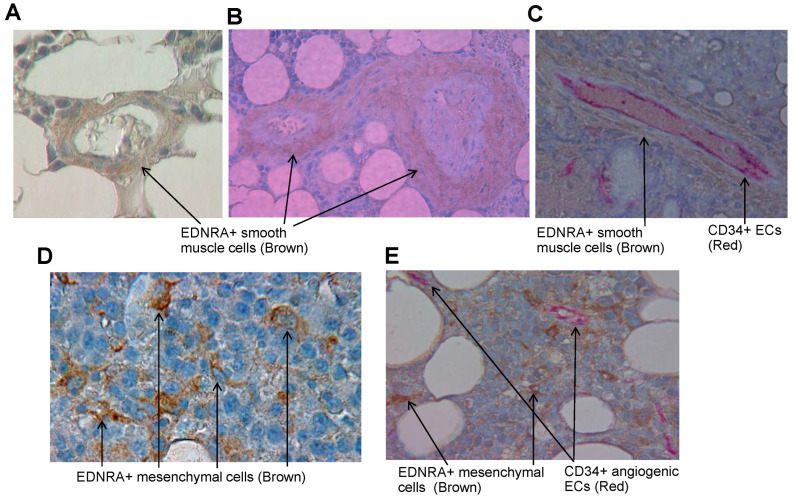
EDNRA is expressed in SMCs of established vessels in normal and myelomatous BM and as detached individual cells in interstitial marrow of patients with MM. (**A**,**B**) A bone section from a healthy donor (**A**) or SMM patient (**B**) showing EDNRA+ SMCs (brown) covering mature vessels. (**C**) Bone sections from an MM patient double stained for EDNRA (brown) and CD34 (red) showing CD34+ ECs in the inner part and SMCs in the outer part of a mature vessel. (**D**) EDNRA+ cells detected in interstitial BM of a patient with MM. Note that EDNRA+ cells have a fibroblast-like shape. (**E**) Double staining of MM bone section for EDNRA (brown) and CD34 (red). CD34+ angiogenic vessels are separated from EDNRA+ cells.

**Figure 4 cancers-15-04519-f004:**
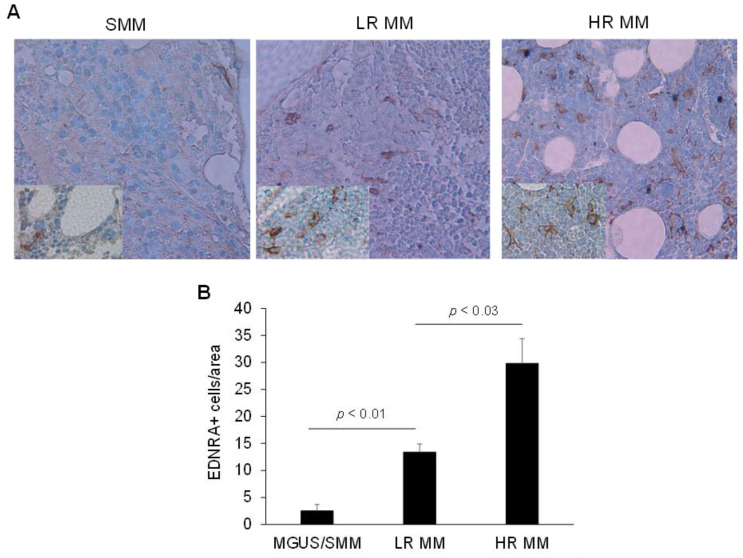
As MM disease progresses, the proportion of EDNRA+ cells in the interstitial BM increases. (**A**) Representative EDNRA staining of bone sections from patients with SMM, low-risk (LR) MM, or high-risk (HR) MM (original magnification, ×20; inserts, ×40). (**B**) Quantification of the number of EDNRA+ cells in the interstitial BM of patients with MGUS/SMM, LR MM, and HR MM. Data expressed as mean ± SEM.

**Figure 5 cancers-15-04519-f005:**
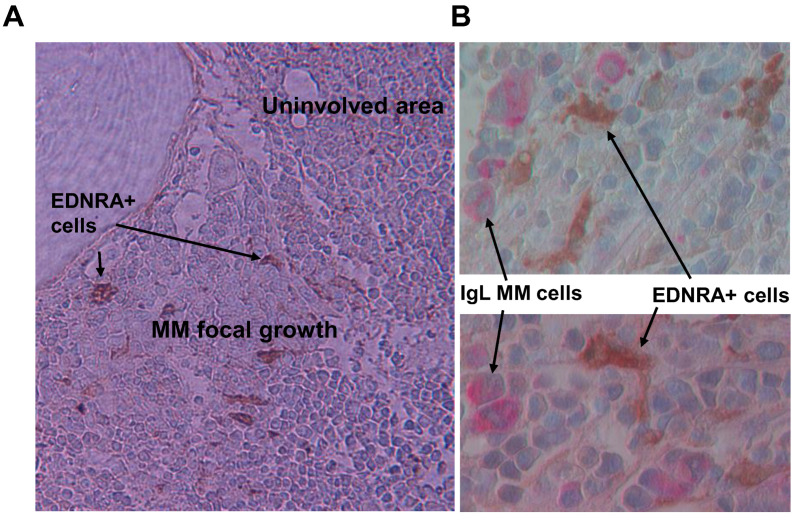
EDNRA+ cells are evident in areas near MM focal lesions. (**A**) IHC staining for EDNRA in bone section from low-risk MM patient. EDNRA+ cells are only present near MM focal growth (×20 original magnification). (**B**) Double staining for Ig lambda (IgL, red) and EDNRA (brown) showing IgL+ MM cells adjacent to EDNRA+ mesenchymal cells (×40 original magnification).

**Figure 6 cancers-15-04519-f006:**
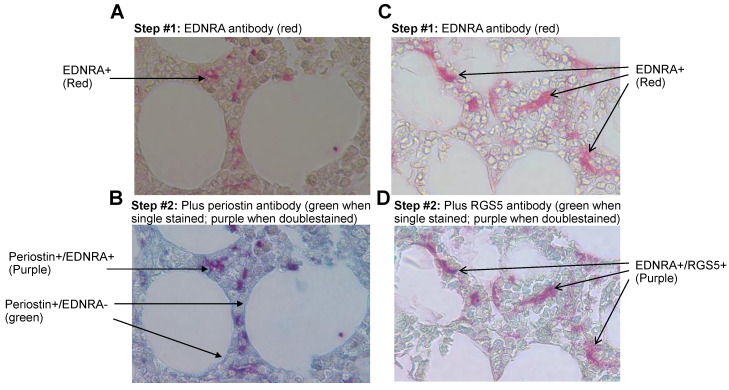
In the interstitial BM of MM patients, EDNRA+ cells coexpress periostin and RGS5. Bone sections were double stained for EDNRA and periostin or RGS5. (**A**,**C**) The first step whereby bone sections were stained for EDNRA (red). (**B**,**D**) Double staining of the same bone sections for periostin (**B**) or RGS5 (**D**) (green) resulted in a purplish color in cells that coexpressed EDNRA and periostin or RGS5.

**Figure 7 cancers-15-04519-f007:**
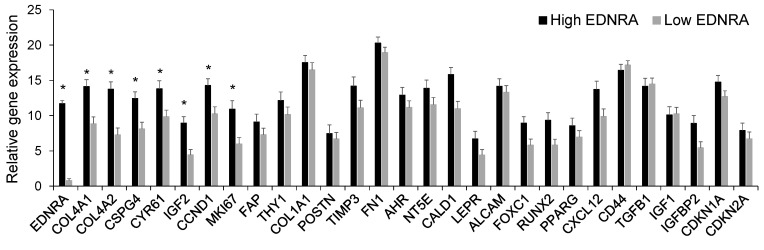
At the single-cell level, cells that express *EDNRA* also express mesenchymal markers and have higher expression of genes associated with proliferation. Unexpanded mesenchymal cells from bone biopsies of a healthy donor and an MM patient were sorted individually and then subjected to qRT-PCR. Individual cells were grouped based on high *EDNRA* expression (*n* = 31) or negative/low *EDNRA* expression (*n* = 60). * *p* < 0.001. Data expressed as mean ± SEM.

## Data Availability

Not applicable.

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
