# Peer review of "EDNRA-Expressing Mesenchymal Cells Are Expanded in Myeloma Interstitial Bone Marrow and Associated with Disease Progression"

_cancers, 2023, doi:10.3390/cancers15184519_

Round 1

Reviewer 1 Report

This is an interesting work. In the present paper the authors show that mesenchymal cells expressing EDNRA are increased in the bone marrow of MM patients, and that the expression of EDNRA increases with disease progression. With immunohistochemistry and single cell PCR, they can show a different localisation of EDNRA expressing cells in disease progression of MM, as well as a different gene expression profile of EDNRA positive cells. I think this work is interesting, as it sets the bases for further work focusing on better understanding the MM microenvironment. It also shows that MSCs are important, and often overlooked, players of MM disease progression, that require more research and attention. The work is nicely done, clearly explained and figures are almost always complete and clear. I do not find any flaws in the experiments performed.

What I miss in this work are some functional experiments, to prove the points of the authors. This is for me the major drawback of the all work.

I do not completely understand figure one. When the authors say EDNRA was lower in MM plasma cells, do they mean sorted plasma cells from patients? Figure legend states CD138 sorted MM plasma cells. Are this sorted plasma cells from the same 521 MM patients on which GEP was performed? In the methods they describe that GEP was performed on whole BM without prior PC separation, and I did not find any description of a separation of plasma cells in the methods. This is important, as the authors claim that EDRA is not expressed in MM cells and is upregulated in the BM microenvironment based on GEP data of unsorted BM and of “MM plasma cells” whose origin is not clear to me.

I think the conclusion has too many suppositions not supported by data from the present paper. When they suggest that EDNRA expressing cells my contribute to MM bone disease or that disruption of normal vascularisation, permits neoangiogenesis and expansion of mesenchymal cells, which may support MM progression, these are just supposition. To prove their point, functional experiments, showing that neoangiogenesis increases the expression of EDNRA, or that for example in mice expressing high EDNRA bone disease is increased should have been performed. As these experiments are lacking, I would suggest reformulating some sentences in the discussion and to focus more to the data presented in the paper.

Reviewer 2 Report

The authors demonstrated that EDNRA+ve mesenchymal cells increase with the progression of MM, and they proposed that these mesenchymal cells might be a novel and valuable biomarker. This viewpoint is fascinating, and these findings are informative for hemato-oncologists. However, the reviewer requests more information described below:

1.         What is the mechanism that EDNRA+ve cells detach from vasculatures and increase by MM progression?

2.         What is the function of these EDNRA+ve cells?

3.         If the authors describe that EDNRA is a biomarker for MM only, they also analyze the correlation of EDNRA+ cells and chemo-resistance and/or the responses to therapy.

4.         Do these EDNRA+ve cells exist in extramedullary MM tumors?

A few typos exist in the text. The authors had better check and rewrite them.

Reviewer 3 Report

1 As we all know, the endothelin-1 (EDN1) axis, consisting of EDN1 acting through EDN-receptor A (EDNRA) and B (EDNRB), was previously shown to be overexpressed in MM. However, there is incomplete understanding of how EDNRA regulates MM growth and response to therapy, they may cause the development of tumors by some pathways involved in cell proliferation, migration, invasion, epithelial-mesenchymal transition, osteogenesis and angiogenesis. The report has found that the EDNRA is overexpressed in whole MM bone biopsies and sought to characterize its expression. EDNRA expression was gradually increased with disease progression. But there is no survival analysis of different expression levels of EDNRA.

2 What is the mechanism behind the increased expression of EDNRA and why EDNRA+ mesenchymal cells in the interstitial BM is overexpressed in high risk MM? Genetic changes or epigenetic effects?

3 In figure 5, it reported that in the interstitial BM of MM patients, EDNRA+ cells coexpress periostin and RGS5, which want to prove the phenotype of EDNRA+ cells in interstitial BM is a marker of pericytes. What is the relationship between EDNRA+ pericytes with MM and what is the function of EDNRA+ cells in the progression of MM?

4 The main method of this article is to use immunohistochemical methods to detect the expression level of EDNRA in cells. The sample sources are all clinical patients. Can the same results be observed in MM cell lines? And in the discussion, the author talked about thalidomide, a frontline anti-MM clinical drug known to inhibit angiogenesis, induces vessel maturation and stability by acting on pericytes and increasing mural cell coverage of the vasculature. Maybe thalidomide can be used in the cell lines of MM to reduce the expression of EDNRA.

Minor editing of English language required, for example the tenses of sentences.

Round 2

Reviewer 2 Report

In their revised manuscript, the authors have adequately addressed all concerns that the reviewers had pointed out.

But, I would like to ask the authors. What is "the intestinal BM ( in line 341)"? Is this mistyping? Is "the interstitial BM (as written in line 317) correct?

Reviewer 3 Report

In their revised manuscript, the authors have adequately addressed all concerns that the reviewers had pointed out.